# Chitosan Ameliorates DSS-Induced Ulcerative Colitis Mice by Enhancing Intestinal Barrier Function and Improving Microflora

**DOI:** 10.3390/ijms20225751

**Published:** 2019-11-15

**Authors:** Jia Wang, Cuili Zhang, Chunmei Guo, Xinli Li

**Affiliations:** Department of Biotechnology, College of Basic Medical Sciences, Dalian Medical University, Dalian 116044, China18340841956@139.com (C.G.)

**Keywords:** chitosan, ulcerative colitis, tight junction protein, intestinal microflora

## Abstract

Ulcerative colitis (UC) has been identified as one of the inflammatory diseases. Intestinal mucosal barrier function and microflora play major roles in UC. Modified-chitosan products have been consumed as effective and safe drugs to treat UC. The present work aimed to investigate the effect of chitosan (CS) on intestinal microflora and intestinal barrier function in dextran sulfate sodium (DSS)-induced UC mice and to explore the underlying mechanisms. KM (Kunming) mice received water/CS (250, 150 mg/kg) for 5 days, and then received 3% DSS for 5 days to induce UC. Subsequently, CS (250, 150 mg/kg) was administered daily for 5 days. Clinical signs, body weight, colon length, and histological changes were recorded. Alterations of intestinal microflora were analyzed by PCR-DGGE, expressions of TNF-α and tight junction proteins were detected by Western blotting. CS showed a significant effect against UC by the increased body weight and colon length, decreased DAI (disease activity index) and histological injury scores, and alleviated histopathological changes. CS reduced the expression of TNF-α, promoted the expressions of tight junction proteins such as claudin-1, occludin, and ZO-1 to maintain the intestinal mucosal barrier function for attenuating UC in mice. Furthermore, *Parabacteroides*, *Blautia*, *Lactobacillus*, and *Prevotella* were dominant organisms in the intestinal tract. *Blautia* and *Lactobacillus* decreased with DSS treatment, but increased obviously with CS treatment. This is the first time that the effect of original CS against UC in mice has been reported and it is through promoting dominant intestinal microflora such as *Blautia*, mitigating intestinal microflora dysbiosis, and regulating the expressions of TNF-α, claudin-1, occludin, and ZO-1. CS can be developed as an effective food and health care product for the prevention and treatment of UC.

## 1. Introduction

Ulcerative colitis (UC) has been identified as one of the modern inflammatory diseases [1]. It is a chronic and nonspecific inflammatory bowel disease (IBD), which presents with abdominal pain, diarrhea and bloody mucopurulent stool [2], and associated with a high risk of colon cancer if not treated in a timely manner.

The pathogenesis of UC is not clearly understood, and it is generally considered to be caused by multiple factors [3]. Among these, intestinal mucosal barrier function and microflora play major roles in which UC occurs and develops [4]. The intestinal mucosal barrier is the first barrier against a hostile environment, mainly formed by the tight junctions (TJs) of epithelial cells. TJs consist of transmembrane proteins (occludins and claudins) and accessory proteins (zonula occludens) for preventing the spread of pathogens and harmful antigens across the epithelium. ZOs (zonula occludens), occludin, and claudins are thought to be important integral membrane proteins which participate in TJ structural integrity by binding to a actin-cytoskeleton [5]. In addition, there is a strong relationship between intestinal microflora and intestinal barrier function. Intestinal microflora dysbiosis decreases the intestinal mucosal barrier function and increases bacterial translocation, and intestinal pathogenic bacteria damage structural barriers by changing intestinal TJ proteins [6]. Thus, the research works focusing on effective methods to regulate intestinal mucosal barrier function and balance intestinal microflora for treatment of UC are important. 

Most drugs for UC treatment interfere with metabolism and immune responses, often causing some serious adverse reactions. Alternative treatments, including probiotics and nutritional supplements, have been given more attention due to fewer side effects [7]. A range of dietary oligosaccharides, such as lactosucrose, galactooligosaccharides, fructooligosaccharides, and isomaltooligosaccharides, have nutritional supplement properties [8,9]. Chitosan (CS) is a polymer obtained by the deacetylation of chitin extracted from the exoskeletons of crustaceans [10], which possesses diverse biological and pharmacological effects, including antitumor, anti-inflammatory, antioxidant, anticoagulant, wound healing, antimicrobial, anti-obesity, and antidiabetic activities [11,12,13]. As a main marine natural compound, CS is a much sought after bioresource due to its therapeutic value, stability, biodegradability, biocompatibility and low toxicity. In recent years, modified-chitosan products have been consumed as effective and safe drugs to treat UC [14,15]. To avoid unintended absorption of the drug or its degradation products in the gastrointestinal tract, rectal administration was used. However, intestinal microflora has an important role in chronic human diseases, it should not be ignored. Our previous work [16] also indicated that CS was a potential food supplement for protecting intestinal microflora. *Lactobacillus* was promoted with a CS treatment in an antibiotics-induced intestinal dysbiosis mice model. In the present study, the curative effect and mechanism of original CS were evaluated in a DSS-induced UC mice model by the alterations of intestinal microflora, and the expressions of TNF-α and tight junction proteins. We attempted to explore the association between intestinal microflora and UC, and to provide a novel insight into the mechanisms of CS. 

## 2. Results

### 2.1. Effects of CS on Body Weight and DAI in DSS-Induced UC Mice

On the first 5 days of the experiment, the mice in each group showed a steady increase in body weight. Subsequently, the mice were treated with 3% DSS for 5 days to induce UC. The control mice still showed a steady increase in body weight. But the DSS-alone treatment group had a significantly decreased body weight and increased DAI score (*p* < 0.01) compared with control mice. Both CSH and CSL treatment groups reduced the body weight loss, and attenuated the increased DAI score (Figure 1A,B). These results indicated that CS effectively relieved DSS-induced UC symptoms.

### 2.2. Effects of CS on Colon Length and Histopathology in DSS-Induced UC Mice

Colon length was shortened in all DSS-treated mice. The colon length of the DSS-only treatment group showed a significant reduction compared with the control group (*p* < 0.01). Both CSH and CSL alleviated the effects of DSS on colon length shortening (Figure 2A,B).

As shown in Figure 2C, intact colonic epithelial cells and crypt structure, and complete goblet cells were observed in the control group. Severe lesions were present in all DSS-treated groups, with loss of colonic epithelial cells, distortion of crypt structure, and massive inflammatory cell infiltration. However, compared with the DSS-only treatment group, the colons of CSH-treated mice showed ameliorated structural damage, exhibited less inflammatory cell infiltration and only mild evidence of crypt distortion. In addition, both CSH and CSL treatment groups resulted in a significant reduction of the histological injury scores caused by DSS (Figure 2D; *p* < 0.01). Taken together, our results suggested that CS significantly protected colon tissue and attenuated DSS-induced tissue morphological changes.

### 2.3. Effects of CS on Expressions of TNF-α, Claudin-1, Occludin, and ZO-1

Western blotting analysis (Figure 3) showed that the expression levels of claudin-1, occludin, and ZO-1 were significantly decreased in the DSS-alone treatment group. Meanwhile, the expression of TNF-α increased significantly in this group. Compared with the DSS-alone treatment group, the expressions of claudin-1, occludin, and ZO-1 increased, but expression of TNF-α decreased in both CSH and CSL treatment groups significantly. Furthermore, as shown in Figure 2D, CS at the dose of 250 mg/kg (CSH) significantly increased the expressions of claudin-1 and occludin, which were even better than the effects produced by CS at the dose of 150 mg/kg (CSL). We proposed that the effects of CS on DSS-induced UC mice were related to the regulation of the colonic mucosal barrier function, where the expressions of ZO-1, occludin, and claudin-1 play important roles in maintaining the intestinal mucosal barrier function.

### 2.4. PCR-DGGE Analysis

The dominant intestinal microflora of experimental groups at different time intervals was examined by PCR-DGGE analysis (Figure 4A). The gene sequencing results were showed in Table 1. Mice pretreated with water/CS for 5 days, and then received 3% DSS for 5 days to induce UC. On the 11th day of the experiment, *Lactobacillus johnsonii* (band d) was found in all groups (D11-Group 1–4). But the intensities of *Blautia* sp. (band c) and *Lactobacillus ruminis* (band e) weakened in the DSS-alone treatment group (D11-Group 2). Especially, *Blautia* sp. (band c) almost disappeared in the DSS-alone treatment group, but this band existed in both CSH and CSL pretreated groups. The intensity of *Blautia* sp. (band c) in the CSH group (D11-Group 3) was stronger than that in CSL group (D11-Group 4). In addition, *Parabacteroides distasonis* (band a) and *Prevotella intermedia* (band f) existed in all DSS-treated groups, but did not exist in the control group. On the 15th day of the experiment, the mice had stopped taking DSS and continued to receive water/CS for 5 days. *Blautia* sp. (band c) still decreased in the DSS-alone treatment group, almost disappearing. *Parabacteroides distasonis* (band a) did not exist in all groups. Other bands had no significant change at different time intervals (D11 and D15). The bacterium from the genera of *Parabacteroides*, *Blautia*, *Lactobacillus*, and *Prevotella* were dominant organisms in the intestinal tract of mice. We further proposed that the alterations of dominant intestinal microflora play causal roles in UC. Specifically, *Blautia* and *Lactobacillus* decreased in the DSS-alone treatment group, which showed that the intestinal microflora balance was disturbed by DSS. CS treatment mitigated intestinal microflora dysbiosis, *Blautia* and *Lactobacillus* increased with CS treatment, and the effect produced by CSH was better than the actions produced by CSL.

Figure 4B displayed that different groups formed the statistically significant UPGMA clustering profiles. There were two main clusters in the dendrogram, the first was D11-Group 2 (related the DSS-alone treatment group (D11)), the second was the remaining seven groups. The minimum bacterial similarity index between cluster one and cluster two was 0.66, which suggested that the intestinal microflora community of DSS-induced UC mice was seriously damaged. In cluster two, there were also two secondary clusters. The maximum bacterial similarity index between D11-Group 1 and D15-Group 1 was 0.83, and D15-Group 2 possessed high similarity (0.76) to the above two groups, which suggested that the intestinal microflora community of normal mice was little changed at D11 and D15. Moreover, the similarity between the CS-treated groups at D11 and D15 was 0.74, which suggested that the treatment cycle of CS had little effect on the intestinal microflora community.

The richness (S), diversity index (*H*′) and evenness score (E) decreased in all DSS-treated groups (Table 2). Compared to the control group, the diversity index was lower in the DSS-only treatment group (*p* < 0.01), but increased in CS-treated groups with no significant difference. It appeared that intestinal microflora community was changed by DSS with S, *H*′, and E decreasing. However, the intestinal microflora community of the mice treated with CS was considerably ameliorated, indicating that CS showed a significant effect on the intestinal microflora in mice.

## 3. Discussions

Increasing evidence [17] has demonstrated that intestinal mucosal barrier dysfunction is critical in UC development. As a key component of the intestinal mucosal barrier, TJ proteins seal the gaps between adjacent intestinal epithelial cells and keeps substances such as antigens and microbes contained in the lumen. They play important roles in the maintenance of intestinal permeability, tissue differentiation, and homeostasis. Previous reports have demonstrated that dietary threonine maintained intestinal barrier function by modulating intestinal TJ proteins synthesis [18]. Fermented *Pueraria lobata* extract ameliorated DSS-induced inflammation in the colon, and recovered the disrupted intestinal barrier through retrieving the expression and architecture of TJ proteins [19]. Expressions of ZO-1, claudin-3, and occludin decreased in enterotoxigenic *Escherichia coli* K88-infected intestinal mucosa damaged pig [20]. Thus, regulation of TJs to keep epithelial barrier integrity in UC is vitally important. Excessive tumor necrosis factor-alpha (TNF-α) expands a local or systemic inflammation, which triggers a disturbance of both TJ proteins and intestinal mucous barrier functions [21], and is associated with UC severity. TNF-α has long been recognized as the key inflammatory mediator in colon inflammation [22]. Blockade of TNF-α activity has proven to be an effective way of inhibiting inflammation. BaweiXileisan (a traditional Chinese compound medicine) inhibited TNF-α expression and improved the mucosa barrier function in DSS-induced UC mice [23]. Hydroxynaphthoquinone mixtures exerted their anti-inflammatory actions through inhibiting TNF-α, down-regulating nuclear factor-κB (NF-κB) signaling. In clinics, TNF-α blockers such as infliximab, adalimumab, and certolizumab pegol have been successfully used for the treatment of IBD patients [24]. In the present study, CS showed significant effects against UC in mice by the increased body weight and colon length, decreased DAI scores and histological injury scores, and alleviated histopathological changes. CS was effective in reducing the expression of TNF-α, and promoting the expressions of TJ proteins such as claudin-1, occludin, and ZO-1 to maintain intestinal mucosal barrier function for attenuating UC in mice, and to this effect, a high dose of CS was better than a low dose.

Probiotics/prebiotics have been suggested as a useful integrative treatment of inflammatory bowel disease, for their ability to alter the expression of epithelial tight junctions. For example, *Lactobacillus rhamnosus* and *Lactobacillus plantarum* strengthened intestinal barrier function, promoted TJ integrity, and protected against experimental necrotizing enterocolitis [25]. A probiotic mixture of *Bacillus subtilis* and *Enterococcus faecium* improved gut microbiota, ameliorated permeability of the intestinal epithelial cell barrier, and enhanced intestinal integrity through up-regulating the expressions of occludin, ZO-1, and JAM-A in heat stress-induced laying hens [26]. Likewise, prebiotic supplements lead to changes in the intestinal microflora, and further improves patients’ well-being and health. Alpinetin decreases intestinal inflammation, and regulates the expression of tight junctions in UC mice. Purple potato extract could be used as a supportive dietary therapeutic strategy for improving gut epithelial health through improving gut epithelial differentiation and barrier function [27]. Accordingly, the effects of probiotic/prebiotics treatment in DSS-induced UC are more important. We were interested in examining whether CS has an impact on the intestinal microflora in UC mice. In the present work, both *Blautia* and *Lactobacillus* decreased with DSS administration for 5 days, and *Blautia* almost disappeared after DSS withdrawal, which showed that the intestinal microflora balance may seriously be disturbed by DSS. However, the CSH group showed higher counts for *Blautia*, and higher diversity index, which demonstrated that a high dose of CS better mitigated intestinal microflora dysbiosis and had beneficial effects on *Blautia*. Additionally, *Blautia* is one of the major intestinal microbes often found in human fecal samples [28]. There is a strong relationship between decreased levels of genus *Blautia* and diseases. Increasing the ratio of *Blautia* in the intestine might be beneficial for health [29]. An increase of *Blautia coccoides* through the intake of Japanese koji might be one mechanism explaining Japanese longevity. So the effect of supplements on health, which increased the ratio of *Blautia* in the intestinal microflora, is of significant concern. Our findings showed that CSH improved *Blautia* and *Lactobacillus*. *Blautia* and *Lactobacillus* can protect the intestine by producing antibacterial substances such as lactic acid and short-chain fatty acids, competing for the nutrients and intestinal adhesion sites to inhibit pathogenic bacteria, and preventing cell apoptosis so as to enhance intestinal barrier function [30]. Moreover, *Blautia coccoides* decreases NF-κB activity in Caco-2 cells [31]. Most anti-inflammatory drugs activate the NF-κB signaling pathway, which in turn promotes and controls the expression of TNF-α and other cytokines. As expected, the expression of TNF-α was reduced with CS treatment. Thus, *Blautia* may act as a key regulator in the pathogenesis of UC.

DSS causes a change in the intestinal microflora composition and induces intestinal barrier dysfunction in mice. Intestinal microflora dysbiosis is further associated with UC and a reversal occurred by CS treatment. We hypothesize that the expressions of TNF-α and TJ proteins may be affected by dominant intestinal microflora such as *Blautia*. CS has prebiotic-like effects which can induce microbial competition and reduce the populations of non-beneficial intestinal microflora. Further studies are required to decipher the role of dominant intestinal microflora in the development of UC.

## 4. Materials and Methods

### 4.1. Material and Reagents

Chitosan (*M*_V_ = 21.70 × 10^4^ Da, DD ≥ 95%) was purchased from Jinan Haidebei Marine Biological Engineering Co., Ltd. (Jinan, China). A voucher specimen (No. CS 201701) was deposited in the Department of Biotechnology, Dalian Medical University, China. DSS (MW 36–50 kD) was purchased from MP Biomedicals (Santa Ana, CA, USA). Stool DNA extract kit was purchased from ForeGenen (Chengdu, China). Polymerase Chain Reaction primers GC-357f (CGCCCGGGGCGCGCCCCGGGCGGGGCGGGGGACGGGGGGCCTACGGGAGGCAGCAG), 518r (ATTACCGCGGCTGCTGG) and 357f (CCTACGGGAGGCAGCAG) were synthesized by TaKaRa Biotechnology Co., Ltd. (Dalian, China). A PCR Mix kit was purchased from Beijing TransGen Biotech Co., Ltd. (Beijing, China). Antibodies against TNF-α, ZO-1, claudin-1, occludin, β-actin, and HRP-conjugated affinipure goat anti-rabbit IgG (H+L) were obtained from Proteintech Group Inc. (Chicago, IL, USA). The enhanced chemiluminescence (ECL) kit was from Amersham Life Science, Inc. (Arlington Heights, IL, USA). All other chemical reagents used were of analytical grade.

### 4.2. Animals and Experimental Design

Male KM mice weighing 20 ± 2 g were provided by the Experimental Animal Center of Dalian Medical University, Dalian, China (Quality certificate number: SCXK (Liao) 2013–0003; 20 May 2013). All experimental procedures were approved by the Animal Care and Use Committee of Dalian Medical University and performed in strict accordance with the People’s Republic of China Legislation Regarding the Use and Care of Laboratory Animals (Approval number: SYXK (Liao) 2013–0006; 18 November 2013). The mice were kept under standardized conditions at a temperature of 22–24 °C, and 20% humidity with a 12 h light/dark cycle, and they had free access to standard diet and water ad libitum. After acclimatization for one week, 40 mice were randomly divided into four groups (*n* = 10). Two groups (Group 1 and Group 2) received normal drinking water only, and the other two groups received CS at doses of 250 (high dose of CS, CSH) or 150 (low dose of CS, CSL) mg/kg by oral gavage for 5 days. On the 6th day, all animals except Group 1 received 3% DSS dissolved in drinking water for 5 days to induce UC [19]. From Day 11 to 15, Group 1 and Group 2 received normal drinking water, and served as control and DSS groups, respectively. The other two groups received CS at doses of 250 or 150 mg/kg by oral gavage. All mice were sacrificed 12 h later after the last administration. The fecal samples were collected on the 11th and 15th day, respectively, and stored at −80 °C for intestinal microflora analysis. The length of the colons was measured and then washed instantly using ice-cold physiological saline. One part of colon tissue was rapidly divided and fixed in 10% formalin for pathological examination, and the remaining parts were stored at −80 °C for Western blotting assay.

### 4.3. Evaluation of Disease Activity Index

The mice were checked daily for UC based on body weight, gross rectal bleeding, and stool consistency. A disease activity index (DAI) score was calculated according to a described method [32] to assess the disease severity.

### 4.4. Colon Histopathology

The length of colon was measured. Then a 0.5 cm colon segment was fixed in 10% formalin for 24 h, paraffin embedded, sliced into 5 μm sections, and stained with hematoxylin-eosin (H & E) for histopathological examination. Each sample was observed at 200× magnification. Histological scores were given on a scale as described previously [33].

### 4.5. Western Blotting Assay

Total protein was extracted from the colon samples using a RIPA lysis buffer with protease inhibitors in a proportion of 1:100. The BCA assay kit was used to quantitate protein. Equal amounts of protein (50 µg) were separated by 10% SDS-PAGE gel using 100 V for 2 h and then transferred to a nitrocellulose membrane by semi-dry apparatus for 40 min for β-actin, 25 min for TNF-α, 30 min for claudin-1, 60 min for occludin, and 180 min for ZO-1, respectively. The membranes were blocked with 5% non-fat milk for 2 h at room temperature and then incubated with primary antibodies against TNF-α, claudin-1, occludin, ZO-1, and β-actin, respectively, at a 1:500 dilution overnight at 4 °C. The next day, the membranes were incubated with secondary antibody at a 1:5000 dilution for 2 h at room temperature after washed with TBST for three times. Then, the protein bands were visualized using an ECL kit by Bio-rad ChemiDoc XRS plus an image analyzer (Bio-Rad, Hercules, CA, USA) after TBST washing, as previously described. β-actin was used as internal reference.

### 4.6. Polymerse Chain Rection (PCR)-Denaturing Gradient Gel Electrophoresis (DGGE) Analysis

Total bacterial DNA was extracted from fecal samples with a Stool DNA kit. For 16S rRNA gene analysis, primers GC-357f and 518r were used to amplify the V3 region. PCR-DGGE analysis was performed with the methods reported in our previous study [34]. Some separated and strong bands were cut out and eluted in 20 μL sterile water at 4 °C overnight. The eluted DNA was reamplified using 357f and 518r primers with the same PCR program. Idiographic sequences were attained by TaKaRa Biotechnology (Dalian, China) Co., Ltd. The results were compared directly with those in GeneBank by Blast search (http://blast.ncbi.nlm.nih.gov/Blast.cgi).

### 4.7. Statistical Analysis

SPSS version 17.0 was used for analysis. *p* values were determined using Student’s *t*-test, *p*-value < 0.01 was considered significant. DGGE gels and Western blotting gels were analyzed by using Quantity One 4.6.2 gel analysis software (Bio-Rad, Hercules, CA, USA). The Shannon–Wiener index of diversity (*H* ′) was used to determine the diversity of the bacterial community. The evenness (E) which reflected uniformity of bacterial species distribution was also computed.

## 5. Conclusions

CS can effectively reduce symptoms in a mouse model of DSS-induced UC and improve intestinal mucosal barrier function and affect intestinal microflora. A high dose of CS better regulates the expressions of TNF-α and TJ proteins such as claudin-1, occludin, and ZO-1. Moreover, the intestinal microflora composition of UC was distinct from controls, and CS treatment can mitigate intestinal microflora dysbiosis. These findings provide novel insights into the mechanisms of CS as a potential agent to ameliorate the severity of UC. Thus, CS can be developed as an effective food and health care product for the prevention of UC and restoration of intestinal microflora balance.

## Figures and Tables

**Figure 1 ijms-20-05751-f001:**
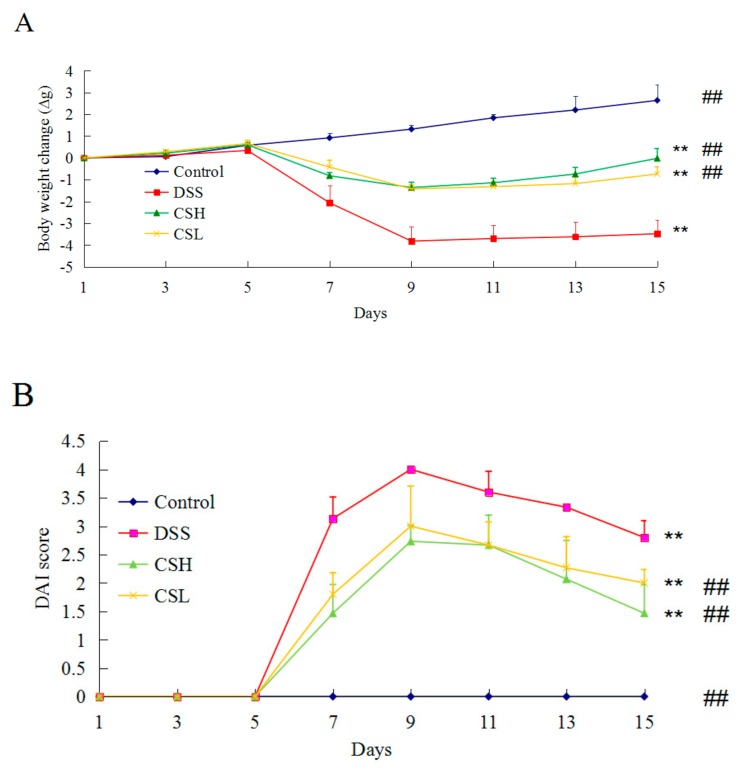
Effects of CS on body weight (**A**) and the disease activity index (DAI) (**B**) in DSS-induced UC mice. Control: normal mice; DSS: mice treated with 3% DSS alone; CSH: mice treated with DSS plus chitosan (250 mg/kg); CSL: mice treated with DSS plus chitosan (150 mg/kg). Values are expressed as mean ± SD (*n* = 10). ** *p* < 0.01 versus control; ^##^
*p* < 0.01 versus DSS-alone.

**Figure 2 ijms-20-05751-f002:**
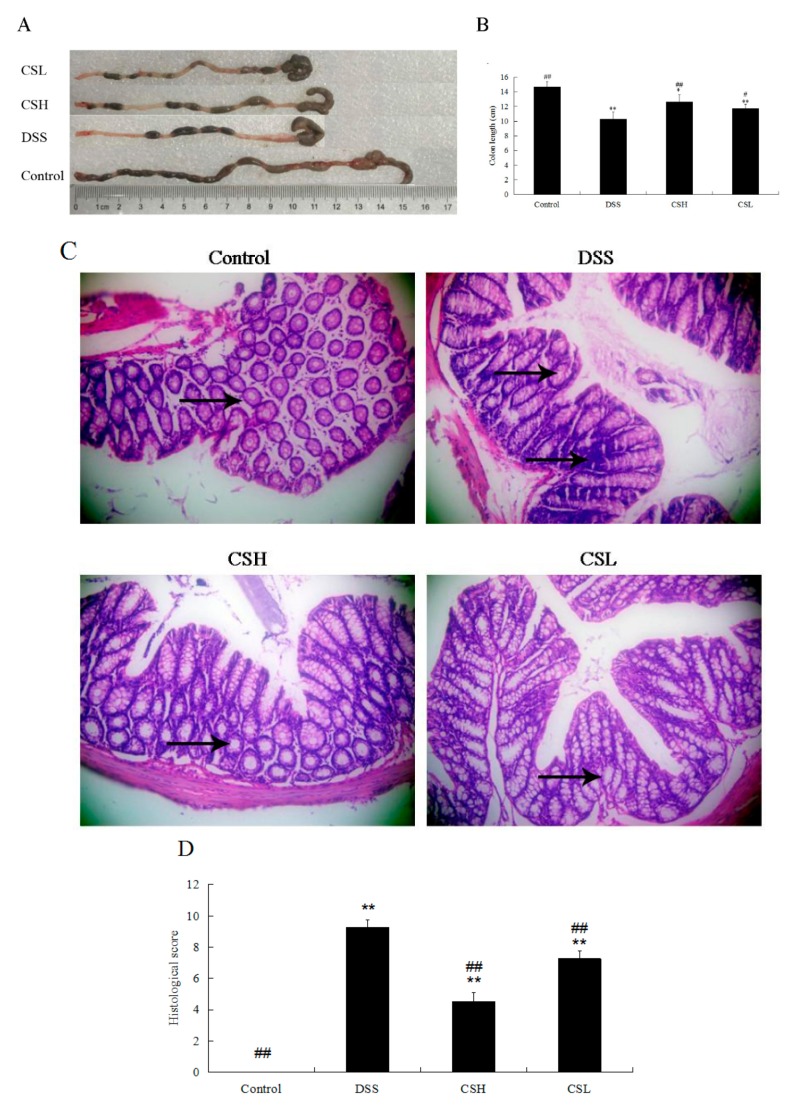
Effects of CS on colon length and histopathology in DSS-induced UC mice. Representative colons (**A**). Colon length (**B**). Histopathology (magnification 200×) (**C**). Histopathological scores (**D**). Arrows indicated the inflammatory infiltration, mucosal erosion, and damage of crypts. Control: normal mice; DSS: mice treated with 3% DSS alone; CSH: mice treated with DSS plus chitosan (250 mg/kg); CSL: mice treated with DSS plus chitosan (150 mg/kg). Values are expressed as mean ± SD (*n* = 10). * *p* < 0.05 and ** *p* < 0.01 versus control; ^#^
*p* < 0.05 and ^##^
*p* < 0.01 versus DSS-alone.

**Figure 3 ijms-20-05751-f003:**
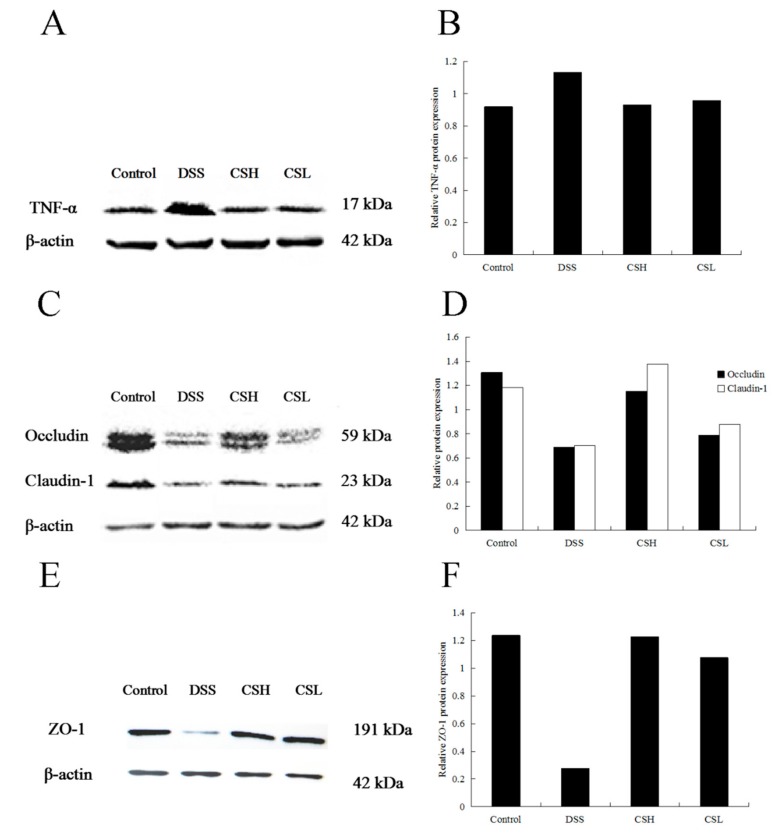
Effects of CS on protein expression of TNF-α, claudin-1, occludin, and ZO-1. Control: normal mice; DSS: mice treated with 3% DSS alone; CSH: mice treated with DSS plus chitosan (250 mg/kg); CSL: mice treated with DSS plus chitosan (150 mg/kg). Values are expressed as mean ± SD (*n* = 10).

**Figure 4 ijms-20-05751-f004:**
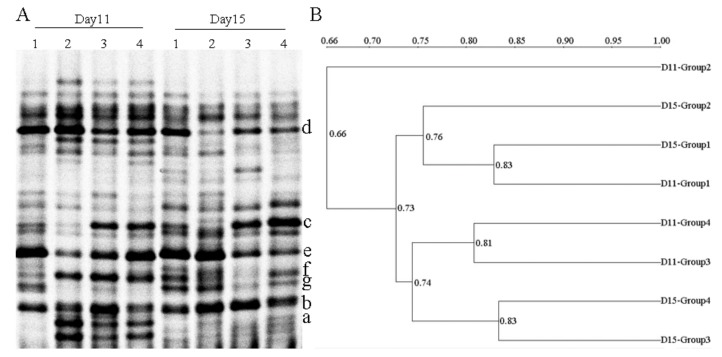
Representative denaturing gradient gel electrophoresis (DGGE) profiles (**A**) and unweighted pair group method using arithmetic average (UPGMA) dendrograms (**B**) of intestinal microflora at different time intervals. D11: the 11th day of the experiment. D11-Group 1: mice treated with normal water; D11-Group 2: mice treated with 3% DSS alone; D11-Group 3: mice treated with DSS plus chitosan (250 mg/kg); D11-Group 4: mice treated with DSS plus chitosan (150 mg/kg). D15: the 15th day of the experiment. D15-Group 1: mice treated with normal water; D15-Group 2: mice treated with normal water; D15-Group 3: mice treated with chitosan (250 mg/kg); D15-Group 4: mice treated with chitosan (150 mg/kg). Bands marked with letters a–g were excised and proceeded for sequencing.

**Table 1 ijms-20-05751-t001:** Sequences of bands a–g based on the BLAST database.

Selected Band	Most Similar Sequence Relative (GenBank Accession Number)	Bacteria Genus	Identity (%)
a	*Parabacteroides distasonis* (NZ 009615.1)	*Parabacteroides*	83
b	*Parabacteroides* sp. (NC CP015402.2)		91
c	*Blautia* sp. (NZ CP015405.2)	*Blautia*	97
d	*Lactobacillus johnsonii* (NZ 022909.1)	*Lactobacillus*	92
e	*Lactobacillus ruminis* (NZ 015975.1)		91
f	*Prevotella intermedia* (NZ CP019301.1)	*Prevotella*	87
g	*Prevotella dentalis* (NZ 019968.1)		89

**Table 2 ijms-20-05751-t002:** Microflora diversity indexes analysis.

Group	S	*H*′	E
D11-Group1	20.50 ± 1.29	3.0679 ± 0.1258	1.0165 ± 0.0439
D11-Group2	18.50 ± 0.58 *	2.7446 ± 0.0469 **	0.9410 ± 0.0259 *
D11-Group3	20.25 ± 0.96	2.9627 ± 0.0473	0.9852 ± 0.0111
D11-Group4	18.75 ± 0.50 *	2.9097 ± 0.0270	0.9929 ± 0.0218
D15-Group1	19.75 ± 0.96	2.9585 ± 0.0480	0.9922 ± 0.0226
D15-Group2	20.25 ± 0.50	2.8867 ± 0.0244 *	0.9597 ± 0.0145 *
D15-Group3	19.25 ± 0.96	2.9241 ± 0.0270	0.9889 ± 0.0267
D15-Group4	20.00 ± 0.82	2.9269 ± 0.0409	0.9775 ± 0.0279

Results are expressed as mean ± SD (*n* = 10). * *p* < 0.05 and ** *p* < 0.01 versus control (D11-Group 1). *H*’ = −∑ (*p_i_*) (ln*p_i_*), *p_i_* was the proportion of the bands in the track, *p_i_* = *n_i_*/∑*n_i_*, *n_i_* was the average density of peak *i* in the densitometric curve. E = *H*’/ln S, S was the number of bands. D11: the 11th day of the experiment. D11-Group 1: mice treated with normal water; D11-Group 2: mice treated with 3% DSS alone; D11-Group 3: mice treated with DSS plus chitosan (250 mg/kg); D11-Group 4: mice treated with DSS plus chitosan (150 mg/kg). D15: the 15th day of the experiment. D15-Group 1: mice treated with normal water; D15-Group 2: mice treated with normal water; D15-Group 3: mice treated with chitosan (250 mg/kg); D15-Group 4: mice treated with chitosan (150 mg/kg).

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
