# Peer review of "Chitosan Ameliorates DSS-Induced Ulcerative Colitis Mice by Enhancing Intestinal Barrier Function and Improving Microflora"

_ijms, 2019, doi:10.3390/ijms20225751_

Round 1
Reviewer 1 Report
The subject of this paper was studied during the last years, the role and use of chitosan in IBD, especially UC, based on its role in intestinal barrier function and microflora.
Before the paper to be published, even that the data presented are interesting and worth to be promoted, some changes should be made, just to name few:
the order of chapters - the results and discussions are presented before the methods; review the first sentence - refractory disease? and the first citation (also in the abstract) use dysbiosis instead of dysbacteriosis to be consistent all over the paper recheck the use of the English language in the paper, better by a native English language speaker review the Figures - for example, Figure 4 legend - maybe lines 155-160 should be included in the legend; better also in the Table 2 legend
Author Response
Dear Glinda He
Assistant Editor (International Journal of Molecular Sciences)
Thank you very much for your kind suggestions on our manuscript.
According to the comments of reviewer 1, we have revised our manuscript as follows:
Question: The subject of this paper was studied during the last years, the role and use of chitosan in IBD, especially UC, based on its role in intestinal barrier function and microflora.
Before the paper to be published, even that the data presented are interesting and worth to be promoted, some changes should be made, just to name few:
the order of chapters - the results and discussions are presented before the methods; review the first sentence - refractory disease? and the first citation (also in the abstract) use dysbiosis instead of dysbacteriosis to be consistent all over the paper recheck the use of the English language in the paper, better by a native English language speaker review the Figures - for example, Figure 4 legend - maybe lines 155-160 should be included in the legend; better also in the Table 2 legend.
Answer: According to the reviewer’s suggestion, we have revised the whole manuscript (red mark).
With best regards.
Yours sincerely,
Xinli Li
E-mail: lixinlibio@hotmail.com
Reviewer 2 Report
Jia Wang et al. aimed investigate effects of chitosan (CS) on intestinal microflora and intestinal barrier function in dextran sulfate sodium (DSS)-induced ulcerative colitis (UC) mice and explore the underlying mechanisms. The authors are reporting that CS showed significant effect against UC by the increased body weight and colon length, decreased DAI (Disease Activity Index) and histological injury scores, and alleviated histopathological changes. They also observed reduced expression of TNF-α, promoted the expressions of tight junction proteins such as claudin-1, occludin, and ZO-1 to maintain the intestinal mucosal barrier function for attenuating UC. They further learnt that parabacteroides, Blautia, Lactobacillus, and Prevotella were dominant organisms in the intestinal tract and that Blautia and Lactobacillus decreased with DSS treatment, but increased with CS treatment.
Clearly, this is the first report on the effect of original CS against DDS-induced UC in mice and the action is through promoting the dominant intestinal microflora mentioned above, mitigating intestinal microflora dysbiosis, and regulating the expressions of TNF-α, claudin-1, occludin, and ZO-1. CS. Despite the fact that there is no drug for UC cure to date, this is plausible attempt. However, the role of CS has been just described in this one paper and its clinical role should be assessed in clinical trials before drawing conclusions. Please note that the results obtained from DDS-induced UC mice experiments may not reflect authentic UC in humans. This is because the inflammation triggered in DDS is actually traumatic and not inflammatory totally different from what is observed in human UC (autoimmune). This paper is very interesting and should probably be rewritten and clearly stated in the title as - “Chitosan ameliorates DDS-induced ulcerative colitis mice by enhancing intestinal barrier function and improving microflora”.
Author Response
Dear Glinda He
Assistant Editor (International Journal of Molecular Sciences)
Thank you very much for your kind suggestions on our manuscript.
According to the comments of reviewer 2, we have revised our manuscript as follows:
Question: However, the role of CS has been just described in this one paper and its clinical role should be assessed in clinical trials before drawing conclusions. Please note that the results obtained from DDS-induced UC mice experiments may not reflect authentic UC in humans. This is because the inflammation triggered in DDS is actually traumatic and not inflammatory totally different from what is observed in human UC (autoimmune). This paper is very interesting and should probably be rewritten and clearly stated in the title as - “Chitosan ameliorates DDS-induced ulcerative colitis mice by enhancing intestinal barrier function and improving microflora”.
Answer: We appreciate the reviewer’s kind suggestion. UC has been identified as one of the modern refractory diseases. Intestinal mucosal barrier function and microflora play major roles in UC. To date, there is no drug for UC cure, we want to do some works to regulate intestinal mucosal barrier function and microflora balance for treatment of UC. This is an attempt, so we have to use DSS to induce a UC mouse model (DSS is often used in the establishment of UC model), and to investigate the effects of CS. We also agree with the reviewer, the results obtained from DDS-induced UC mice experiments may not reflect authentic UC in humans. So we have revised the whole manuscript (red mark) and changed the title to - “Chitosan ameliorates DDS-induced ulcerative colitis mice by enhancing intestinal barrier function and improving microflora” to avoid ambiguity.
With best regards.
Yours sincerely,
Xinli Li
E-mail: lixinlibio@hotmail.com
Round 2
Reviewer 1 Report
I would change also the "refractory disease" and use another word - I do not find in the paper cited [1] anything about UC as a modern refractory disease. This recommendation was not followed.
Otherwise, the results are interesting and worth to be published.
Author Response
Dear Glinda He
Assistant Editor (International Journal of Molecular Sciences)
Thank you very much for your kind suggestions on our manuscript.
According to the comments of reviewer 1, we have revised our manuscript as follows:
Question: I would change also the "refractory disease" and use another word - I do not find in the paper cited [1] anything about UC as a modern refractory disease. This recommendation was not followed.
Answer: According to the reviewer’s suggestion, we have rechecked the reference [1], and changed <refractory diseases> to <inflammatory diseases> (red mark).
With best regards.
Yours sincerely,
Xinli Li
E-mail: lixinlibio@hotmail.com